# Application of Sensitivity Analysis to Discover Potential Molecular Drug Targets

**DOI:** 10.3390/ijms23126604

**Published:** 2022-06-13

**Authors:** Malgorzata Kardynska, Jaroslaw Smieja, Pawel Paszek, Krzysztof Puszynski

**Affiliations:** 1Department of Biosensors and Processing of Biomedical Signals, Silesian University of Technology, 41-800 Zabrze, Poland; malgorzata.kardynska@polsl.pl; 2Department of Systems Biology and Engineering, Silesian University of Technology, 44-100 Gliwice, Poland; krzysztof.puszynski@polsl.pl; 3School of Biology, Faculty of Biology, Medicine and Health, University of Manchester, Manchester Academic Health Science Centre, Manchester M13 9PT, UK; pawel.paszek@manchester.ac.uk

**Keywords:** bioinformatics, chemotherapy, sensitivity analysis, molecular drug targets, systems biology

## Abstract

Mathematical modeling of signaling pathways and regulatory networks has been supporting experimental research for some time now. Sensitivity analysis, aimed at finding model parameters whose changes yield significantly altered cellular responses, is an important part of modeling work. However, sensitivity methods are often directly transplanted from analysis of technical systems, and thus, they may not serve the purposes of analysis of biological systems. This paper presents a novel sensitivity analysis method that is particularly suited to the task of searching for potential molecular drug targets in signaling pathways. Using two sample models of pathways, p53/Mdm2 regulatory module and IFN-β-induced JAK/STAT signaling pathway, we show that the method leads to biologically relevant conclusions, identifying processes suitable for targeted pharmacological inhibition, represented by the reduction of kinetic parameter values. That, in turn, facilitates subsequent search for active drug components.

## 1. Introduction

Computational models of dynamics of intracellular processes have been extensively used in recent years as a support tool in unveiling the structure of regulatory networks [1,2], crosstalk between signaling pathways [3,4], analysis of therapeutic drug effects [5,6,7] or in studies focused on general properties of such systems [8,9,10]. They facilitate preliminary testing of biological hypotheses and provide insights into systems that, for various reasons, cannot be explored experimentally.

For several years, sensitivity analysis have become an ubiquitous tool in the analysis of the mathematical models. In general, sensitivity analysis allows to study the impact of model inputs on its outputs. This work focuses only on examining the impact of model parameters changes on its response. Various techniques have been developed, from local approaches with sensitivity functions [11,12] or sloppy/stiff analysis [13,14,15], to global, variance-based methods [16]. Though some of the papers utilized sensitivity analysis in the context of potential drug targets [17], the methods employed there did not take into account the relation between drug action and kinetic parameters changes. The desired therapeutic actions may consist of either increasing or decreasing the kinetic rates of the processes that are targeted. In mathematical models, this would be reflected in increasing or decreasing corresponding parameters, respectively.

In this paper, we present a novel method of sensitivity analysis that is specifically tailored to the goal of finding potential molecular drug targets. It is based on the analysis of models, given in the form of ordinary differential equations, describing dynamics of responses of a signaling pathway of interest. It allows to study the drug-induced changes in model responses by introducing changes in parameter values, representing effects of drug action. Furthermore, the method makes it possible to tackle the problem of heterogeneity of cellular responses to a drug in a cell population [18,19], using a particular parameter randomization procedure, tailored to the model application.

The proposed method belongs to the family of one-at-a-time (OAT) sensitivity methods, which are based on changing a single model parameter while retaining nominal values of remaining parameters. In general, such approaches are not recommended for nonlinear models or those with interactions between inputs [20]. However, it should be emphasized that the purpose of this work is not to investigate model sensitivity in general, but to utilize sensitivity analysis methods for a precisely defined aim—in the search for potential molecular drug targets. The idea is to find a single parameter whose change significantly alters a single intracellular process. In mathematical models describing biochemical reactions, each parameter is related to a single biochemical process, or the sequence of processes simplified to a single-step process, and reflect the kinetic properties of the molecules (e.g., enzymes) involved in this process [21]. The creation of a parameter ranking facilitates finding a process, the alteration of which, e.g., by drug actions, will lead to significant changes in cellular responses to a given stimuli [12]. Assuming that parameters are independent from each other and that the drug selectively binds to a single target molecule (and thus, altering the kinetic rate of the process in which the molecule participates), it is the OAT approach that should be used.

The highest-ranking parameters lead to the processes that should be targeted in order to make the highest impact on the system responses. The subsequent identification of molecules involved in these processes provides information about potential molecular drug targets. This, in turn, could be the preliminary step before a particular active drug molecule is searched for, using the methods reviewed in, e.g., [22]. On the other hand, the proposed approach might be considered as complementary to network-based methods, also reviewed in [22].

## 2. Results

In order to check the feasibility of the proposed method, two models of different signaling pathways were analyzed: a p53/Mdm2 signaling pathway and an IFN-β-induced JAK/STAT signaling pathway. They represent different dynamical properties (the time responses of the p53 model are oscillatory, whereas in the IFN-β-induced JAK/STAT model the time responses are aperiodic) and different complexities (the p53 model contains much fewer variables than the second one, but its nonlinearities are more complex). The detailed equations and nominal parameters are not listed in this paper, as they have been implemented exactly in the form described in the literature.

The results obtained using the described method were compared with the results of a classic, local sensitivity analysis using the sensitivity functions with parameter rankings based on the area under the curve of the sensitivity function [12,23].

### 2.1. p53 Regulatory Module

The first example involves the regulatory module controlling the level of tumor protein p53 in a cell. p53, often called the Guardian of the Genome [24], is involved in many intracellular processes, among which the most important is cell cycle arrest, expression of DNA repair proteins and induction of cell apoptosis in response to moderate and strong stress signals [25,26,27]. The activation of the p53 signaling pathway occurs in response to environmental stress factors that cause DNA damage or mutations (e.g., ionizing radiation [28]). p53 protein dysfunctions are of great importance in cancer progression. It has been shown that about half of the cancer types have mutations in the p53 gene (TP53), while in many others, malfunctions of other proteins involved in the p53 signaling pathway are observed [29]. Therefore, a search for a potential drug target might be based on the primary goal of inducing a high level of p53 in cells, ultimately leading to their death [19]. Moreover, the results of the sensitivity analysis can be related to the existing research on anticancer drugs targeting the p53 system [30,31].

We chose a relatively simple model of the p53 signaling pathway [32] described by 12 differential equations and 43 parameters (see the Appendix A for the equations). The model describes the negative feedback loop between p53 protein and its inhibitor—the murine double minute 2 (Mdm2) protein, as well as positive feedback loop, in which p53 via phosphatase and tensin homolog deleted on chromosome ten (PTEN), phosphatidylinositol (3,4,5)-trisphosphate (PIP3) and Protein kinase B (Akt) inhibits its own Mdm2 inhibitor. The interactions occurring in this model are shown in the diagram (Figure 1). The level of phosphorylated p53 ($p53_{PN}$) protein has been chosen as the variable, whose sensitivity to the parameter changes represents the sensitivity of the whole system.

Out of the 38 parameters in the model, 3 describe constants such as the Michaelis Menten coefficients or saturation constants and were assumed to take the nominal values and not considered in sensitivity analysis of them ($h_{0}$, NSAT, $d_{DAM}$). The sensitivity ranking for the remaining 35 parameters is presented in the lower panel of Figure 2 and compared with one of the standard rankings based on sensitivity functions in the upper panel of Figure 2. Parameter names corresponding to the numbers in the ranking are presented in Table 1.

Assuming the apoptosis-promoting character of the potential drug for which the molecular targets would be searched for, the parameters of interest should be located above the $10^{0}$ threshold line, as indicated in the closing paragraph of the Materials and Methods section. Then, their increase would lead to prolonged elevated levels of nuclear p53.

The ranking, shown in Figure 2 (bottom panel) apparently indicates different parameters than those drawn from a sensitivity functions-based one (Figure 2 (upper panel)). It suggests that the system response would be affected most efficiently by decreasing one of the parameters represented by the following numbers: 7, 8, 9, 15, 17, 21, 27, 28, 30 and 31. For all these parameters, we observe similar, high values of the A index (Figure 2 (bottom panel)). Having checked their meaning (Table 1), we found that parameters 7, 8, 21, 27, 30 and 31 are associated with processes related to proteins PTEN, PIP3 and AKT, which are also involved in regulation of other intracellular processes, not associated directly with apoptosis (e.g., mTOR or glucagon pathways). Therefore, we decided to leave them out of further analysis.

Of the remaining four parameters, we chose parameters no. 15 and 17, representing Mdm2 transcription rate and Mdm2 translation rate ($s_{0}$ and $t_{0}$—see Table 1) for further analysis. To verify the conclusion drawn from the ranking, a simulation was run for a nominal parameters and mean of the reduced parameter $s_{0}$ ($0.15s_{0}$) and $t_{0}$ ($0.15t_{0}$), respectively.

The results show that the time course of the p53 level in a cell would be substantially altered, following the parameter value reduction (Figure 3), leading to elevated p53 levels that could, ultimately, result in cell apoptosis. In all simulations, obtained after changing the top-ranking parameter values, the up-regulation of p53 is persistent and levels of p53 should be sufficient to induce desired cellular response, but in the case of the reduction of parameters $s_{0}$ and $t_{0}$, it is substantially stronger and appears earlier.

Though the response is shown for a single cell only, the high ranking of the chosen parameters means that a similar response would be observed in most cells in a population (Figure 4), due to the structure of the algorithm and the distribution of the reduction factor α that was used for ranking calculation.

Contrary to our method, a ranking based on sensitivity functions suggested that the $d_{rep}$ parameter (DNA repair rate) should be more important than the parameters associated with the Mdm2 transcription and translation ($s_{0}$ and $t_{0}$). Figure 5 shows that the reduction of the $d_{rep}$ parameter altered the p53 protein response; however, a substantially higher level of p53 was obtained after reducing parameters $s_{0}$ or $t_{0}$.

Moreover, some parameters, e.g., $d_{2}$, $d_{4}$ and $d_{6}$, are low-ranked in the sensitivity function-based ranking (see the upper panel in Figure 2), and as such, would not be considered for further investigations, if that method was applied. However, Figure 6 clearly shows that these parameters should be ranked higher. This example demonstrates that classical sensitivity analysis methods based on sensitivity functions are not suitable for searching for molecular targets for new drugs.

### 2.2. IFN-β-Induced JAK/STAT Signaling Pathway

As the second example, the model of Interferon-β (IFN-β) induced Janus kinase/signal transducer and activator of transcription (JAK/STAT) signaling pathway has been considered. It plays a critical role in the pro-inflammatory immune responses to viral infections [33,34]. The IFN-β cytokine is used in treatment of many diseases, including multiple sclerosis and cancer (see, e.g., [35,36,37]), as well as in viral infections [38,39].

The model from [1] has been chosen (Figure 7). Its temporal response, contrary to the previous example, is not oscillatory. The model contains 27 variables and 50 parameters (see the Appendix A for the equations). Interferon regulatory factor 1 (IRF1) mRNA concentration has been selected as the variable representing system behavior, whose sensitivity to parameter changes is analyzed. For a detailed description of equations and the meaning of parameters, see [1].

Two parameters of the model were omitted in the sensitivity analysis: $k_{s1\_phos\_sat}$ and $k_{s2\_phos\_sat}$. They represent saturation constants and, as such, would not provide information about a prospective molecular drug target. The resulting parameter rankings are shown in Figure 8. Parameter names corresponding to the numbers on the ranking can be read from Table 2.

Interestingly, in the case of the IFN-β-induced JAK/STAT signaling pathway, both rankings indicate the same parameters as the most important for the pathway temporal response, which may result from a smaller number of nonlinear terms in the IFN-β-induced JAK/STAT signaling pathway model. However, the method proposed in this paper allows to discern parameters that inhibit the pathway response from those that amplify it.

If the desired drug action was to amplify the pathway response, in order to strengthen the immune response, then the ranking suggests that any of the parameters below the red threshold line in Figure 8 (bottom panel) should be considered for further analysis. The meaning of the mentioned parameters can be found in Table 2. Of these, parameters 23, 28 and 30 correspond to processes involving a hypothetical pathway-activated phosphatase (PHY), suggested in [1] and as such could not be used in drug search, unless it is identified in experimental research. The parameters 22 and 47 are related to processes involving the STAT2 protein, which is important for other processes of immune response, and changing them would amplify these, in turn. Therefore, we focused on parameter 16 ($k_{i1t\_deg}$ in the original model), which is the degradation rate constant of the IRF1 mRNA. The result of decreasing its value is shown in Figure 9 with a dotted line.

If the desired drug action was to inhibit the pathway response, e.g., in autoimmune diseases, p45 ($k_{s1\_phos}$ in the original model—STAT1 phosphorylation rate) would be the parameter of choice, according to the ranking shown in Figure 8. Reduction of its value would lead to almost complete inhibition of the system response (Figure 9, dashed line).

## 3. Discussion

In both examples introduced in the preceding section, sensitivity rankings provided information about important parameters, associated with particular processes to be targeted by a potential drug. The direct effect of the drug is assumed to be the reduction of kinetic rates of these processes. Such reduction might be achieved, e.g., by binding of an active drug component to one of the molecules involved in the given process. With that in mind, a search for such component might be initiated with molecular dynamics methods.

In the first model, two parameters, $t_{0}$ and $s_{0}$, were the most important in the model. These two parameters are associated with processes leading to Mdm2 protein production (the former one indirectly, through the production of Mdm2 mRNA). Since the translation process may be targeted by specifically designed siRNA, the parameter $t_{0}$, representing the Mdm2 translation rate, seems to indicate a promising candidate for a molecular drug target. According to the ranking results, devising siRNA that would target Mdm2 mRNA would significantly alter system responses, which was already proposed in another in silico study [40]. There are also other high-ranking parameters, and each of them should be looked at from biological point of view. For example, the parameters $d_{4}$ and $d_{6}$ (no. 23 and 25, respectively, in Figure 2), representing the Mdm2-induced degradation rates of p53 and phospho-p53, respectively, might be another parameters of interest. In fact, nutlin, a drug targeting that Mdm2-induced degradation of p53 has been clinically tested [7,41]. Since nutlin targets Mdm2 and, thus, affects p53 degradation regardless of p53 form, we have also checked the impact of simultaneous reduction of $d_{4}$ and $d_{6}$, representing nutlin actions (Figure 10). The corresponding level of p53 induction is lower than the one exhibited by changing parameters $t_{0}$ or $s_{0}$, but still sufficient to induce cell apoptosis.

The ranking obtained in the second example shows that, depending on the desired drug action, molecular targeting should concentrate on decreasing of either the degradation rate $k_{i1t\_deg}$ of the IRF1 transcript (increasing its half-life) or phosphorylation rate $k_{s1\_phos}$ of STAT1 proteins. In the first case, the aim would be to prolong the cellular response to the pathway activation through stabilizing IRF1 mRNA. Although at the moment there are no available means of increasing IRF1 transcript half-life, this might change in the future, e.g., through the use of microRNAs (miRNAs) and molecules targeted at miRNAs (antimiRs) [42] or other RNA drugs-oriented methods (reviewed recently in [43]). In the second case, the decrease of the STAT1 phosphorylation rate would be important for attenuating inflammatory responses. This is particularly interesting, as it would mean restoring the SOCS-1-mediated regulatory mechanism [44,45,46], not present in HeLa cells, for which the mathematical model of the pathway was developed [1]. Therefore, inhibiting STAT1 phosphorylation by means of an active drug component which acts in an SOCS-1-like manner, could prevent prolonged inflammation.

## 4. Materials and Methods

### 4.1. The General Class of Models Considered

There are many methods of modeling signaling pathways and regulatory networks [22,47,48,49]. The approach presented in this paper assumes that the mathematical model is given in the form of ordinary differential equations, describing changes in concentrations of molecular species involved in the intracellular processes under investigation.

In general, these equations may be written as follows: (1)$$\frac{dX}{dt}=F\left(X,P,u\right)$$
where $X={\left[X_{1}\;\ldots \;X_{i}\;\ldots \;X_{n}\right]}^{T}$, $P={\left[p_{1}\;\ldots \;p_{i}\;\ldots \;p_{k}\right]}^{T}$, with $X_{i}$ and $p_{i}$ denoting concentration or number of molecules of type *i* and model parameters, respectively. The vector u, which is a control vector in systems theory, represents external or internal excitation of the system. It is worth noting that the function F does not have to be continuous and may contain stochastic switches representing random signaling events [6].

Let the solution of the model (Equation 1) be given by: (2)$$X=X(P_{\mathrm{nom}},t,u),$$
where $P_{\mathrm{nom}}$ denotes the nominal parameter vector. Nominal values of parameters of the model (Equation 1) are estimated basing on the results of biological experiments, in procedures that provide the best fit to experimental data. The general goal of sensitivity analysis is to find parameters whose change affects the solution (Equation 2) the most. In order to do that, a rank of each parameter is established, according to the value of the sensitivity index, defined by the method used in the analysis. For complex systems, building such rankings requires introduction of some kind of a combined ranking index, as the extent to which each of the variables $x_{i}$ is affected by a parameters change may vary. However, when the analysis is focused on regulatory networks or signaling pathways, usually, one variable is chosen to be representative of the system, thus simplifying the task. That variable denotes the concentration of a protein, transcript, protein complex or other key molecule that, e.g., defines the fate of the system, such as the p53 protein in the example that is included in this paper.

### 4.2. Sensitivity Rankings

The solution of the model (Equation 1) is a function of time. Depending on a goal the model has been developed for, one may consider one of different response characteristics to be the focal point of the sensitivity analysis. These include, among others, steady state value, time needed to reach the steady state, the dominant oscillation frequency, frequency spectrum of the system transient response [50,51,52] or the transient time response. Though in many cases, steady state is the preferred system characteristic, in general, it would be unsuitable for cases when it is the transient response, not the steady state, that is affected by the parameter change. The simplest example would be a temporary increase of a protein level, followed by fallback to its initial condition. Similarly, a dominant oscillation frequency would not be a proper choice for aperiodic systems. Moreover, when the study is focused on finding potential drug targets, it seems that quantitative changes are more important than qualitative ones, and therefore, in this paper, we propose to base the sensitivity indices on the area under the curve (AUC) of the model time response.

Following the assumption given in the Introduction section, in the approach presented below, one of the state variables is arbitrarily chosen to represent the system output, due to its importance both in the particular pathway/regulatory module under investigation and in other cell responses. That simplifies the ranking construction and seems to be acceptable, at least in the examples that follow.

The difference between the system responses $x_{i}$ obtained for nominal and modified parameter $p_{j}$ is given by: (3)$$\Delta x_{i}\left(p_{j},t\right)=x_{i}\left(p_{j},t\right)-x_{i,nom}\left(p_{\mathrm{nom}},t\right).$$

The following integral can be used as a measure of the effect of parameter change over the time horizon *T*, forming a foundation for the parameter ranking: (4)$$R_{j}=\int_{0}^{T}\Delta x_{i}\left(p_{j},t\right)dt$$

Usually, such an integral is calculated of either ${\left(\Delta x\right)}^{2}$ or |(Δx)|. However, taking into account the goal of the analysis, significant parameters should yield a Δx that is of the same sign over the time horizon *T*. Therefore, using (Equation 4) is acceptable and yields additional information about the type of the parameter impact—the negative value is related to the inhibition drug action, whereas a positive value is related to the drug-induced amplification of the system response.

A molecular drug may either increase or decrease the ratio of a biochemical process in a pathway of interest. The respective kinetic parameter should be substantially increased or reduced, correspondingly (by the so-called amplification or reduction factor) and the impact of such change measured. Since individual cells may be affected to a different degree by the drug [18,19], the alteration factor should be a random number. That way, heterogeneity of cellular responses to a drug in a cell population is incorporated into analysis.

The basic algorithm, proposed in this paper involves the following steps:
Run the simulation for nominal parameter values $p_{\mathrm{nom}}$, obtaining $X_{\mathrm{nom}}\left(p_{\mathrm{nom}},t\right)$, and select a state variable representing the system output $x_{i,nom}\left(p_{\mathrm{nom}},t\right)$.For each parameter $p_{j}$, generate a large set of its random values $\left[\alpha \cdot p_{j,nom}\right]$, where α is an alteration (amplification/reduction) factor drawn from a chosen distribution with mean μ equal to the the average drug effect in the population of cells and standard deviation σ representing a heterogeneous response to this drug.For each generated parameter value $p_{j}$:Run a simulation with randomized parameter $p_{j}$ and remaining parameters at their nominal values, obtaining $x_{i}\left(p_{j},t\right)$;Calculate the difference between the nominal response and the new response, defined by (Equation 3);Calculate the sensitivity index:
(5)$$S_{j}=\frac{\int_{0}^{T}\Delta x_{i}\left(p_{j},t\right)dt}{\int_{0}^{T}x_{i,nom}\left(p_{\mathrm{nom}},t\right)dt}$$
where T is the end time of the simulation.Calculate the mean $\bar{\mu }$ of the index *S* for each parameter:
(6)$$A_{j}=\bar{\mu }\left(S_{j}\right).$$Create a parameter ranking, where each parameter $p_{j}$ has been assigned the value $A_{j}$.

The general idea of the proposed algorithm is illustrated in Figure 11.

As mentioned in the preceding section, we concentrate on inhibitory drug actions. For example, when binding to their respective targets, a drug may block the active site that becomes unavailable for other molecules, and thus, makes the target unavailable for a given process. By blocking the active sites of selected molecules, it is also possible to indirectly stimulate the process, e.g., through blocking the inhibitor of that process [7]. The method shown in the paper is general enough also in the case when stimulation is straightforward, as is in the case of recombinant proteins [53]. The only difference from the case studies shown in the Results section would be in the distribution used for sampling α values that should be the amplification factor.

The inhibitory drug actions are represented in a model by parameter reduction. The choice of a range, from which the reduction factor α should be sampled is based on the following assumptions:If the potential drug is to be effective, it must substantially reduce the kinetic parameter—by 85% or more in most cells;Some of the cells may be partially or totally resistant to the drug.

The latter assumption would not be satisfied, if a uniform distribution was used for sampling α, as suggested in [54].

The reduction factor α, representing the drug inhibitory potential, was drawn from the log-normal distribution with μ=−2.08 and σ=0.61. Log-normal distribution parameters were chosen to obtain an average value of the factor α equal to 0.15 (85% parameter reduction). The distribution of the reduction factor α is shown in Figure 12.

However, drug impact might be brought by an increase of parameter values, instead of their decrease. This is the case, e.g., in enhancing signal transmission by the drug retigabine [55,56]. Another example can be found in Michelis–Menten-type kinetics, with the process kinetic ratio given by $\frac{kx(t)}{k_{m}+x\left(t\right)}$, where the drug targeting an enzyme that facilitates the process would lead to increasing the value of $k_{m}$. Then, the only change in the algorithm would involve sampling parameter α>>1 (which would become an amplification, not a reduction factor) in step 3 of the proposed algorithm. In the case of reducing parameter values, the sampling described in the preceding paragraph may be considered to be general, regardless of the process associated with a parameter. However, when increasing parameters, the choice of a range parameter sampling would have to be determined separately for each specific case. Expert opinions on the feasible maximum amplification factor would be needed. Therefore, in the examples shown in the subsequent section, we constrain ourselves to the examples which consider potential drug-induced reduction of parameter values only.

Negative *S* values, resulting from negative $\Delta x_{i}\left(p_{j},t\right)$, should be interpreted as the suppression of the model response after changing the parameter, and the minimum *S* value may be equal to −1, which represents the complete suppression of the model response (i.e., after changing the parameter value, the variable $x_{i}$, representing the level of i-th molecules, has the value 0 over the entire simulation). Positive values of *S* correspond to the elevated level of the *i*-th molecules, and the maximum value of *S* is not limited. Therefore, if it is the parameter reduction-oriented analysis, it seems reasonable to present the parameters ranking on a logarithmic scale (with values of *A* scaled up by adding 1 to avoid negative arguments in log function). Then, in the rankings presented in this paper, values above $10^{0}$ correspond to the amplification of the model response, while values below $10^{0}$ correspond to the suppression of the model response. In other words, if the goal is to inhibit a signal in a pathway, parameters much below that threshold are of interest (the lower the value, the better). Otherwise, if drug action should yield a response amplification, the highest-located points represent parameters of interest.

Simulations were run in MATLAB R2020b using the ode23tb function.

## 5. Conclusions

The method of parameter ranking creation that has been introduced in this paper allows to find parameters most sensitive to targeted inhibition in signaling pathways and regulatory network models. It can be applied to a model of any pathway, regardless of its size, complexity and dynamical properties, as long as it is described by ordinary differential equations. The examples included in this paper prove that the conclusions drawn from the rankings are biologically relevant. It should be noted that the method takes into account heterogeneity in cell populations and allows to find prospective targets even if some cells ultimately appear to be not responding to treatment. The range of such heterogeneity is defined by two parameters of the distribution used to sample the reduction factor α in the model.

The approach presented in the paper may also be used in analysis focused on drugs amplifying cellular responses as represented by kinetic ratio increase. That, however, requires expert knowledge of a pathway under consideration at the time of simulations setup, as only parameters whose increase would be biologically relevant should be tested in the computational procedures.

Once the most important parameters and the processes they are associated with are identified, the next step would be to employ bioinformatics algorithms and molecular modeling [57,58] to find prospective drug components that would bind to the molecules involved in these processes, thus completing in silico search for a potential drug and preparing the stage for biological experiments.

Though the methods and examples introduced in this paper are very promising, one should remember that the models analyzed describe only a selected part of a very complex system of regulatory networks regulating intracellular processes. Such models are always based on an assumption that processes not included do not affect significantly the behavior of the system under investigation (at least, over the time horizon that is used in the simulation study). Moreover, responses to the given stimuli may be cell-type or tissue-specific, and a model that provides a good fit to experimental data in one experimental setup may need to be redesigned for another one. Therefore, in silico findings may not necessarily apply in biological in vitro or in vivo systems. Additionally, there may be off-target effects; inhibiting one target may affect other regulatory networks, and while the proposed approach is useful in search for prospective drug targets, one should be aware of its limitations, which can be summarized as follows:A drug may affect several parameters simultaneously and drive the parameters change in the same direction [59].Results of the analysis depend on the structure of the model and nominal parameter values. As a consequence, a model with estimated parameters that fits experimental data is needed, possibly one that takes into account, e.g., mutant proteins and their effects on the regulatory network.It is not necessary that in silico findings will apply in biological in vitro and in vivo systems.Signaling pathway models describe only a small fraction of much larger systems. The drug may also affect other signaling pathways (unforeseen side effects of drugs)—bioinformatics analysis is needed and expert knowledge is required to confirm the biomedical relevance of the findings.

Nevertheless, despite the limitations listed above, the approach proposed in this paper should facilitate faster advances in search for new molecular drug targets, with its ability to indicate where to look for the most efficient way of affecting cellular responses.

## Figures and Tables

**Figure 1 ijms-23-06604-f001:**
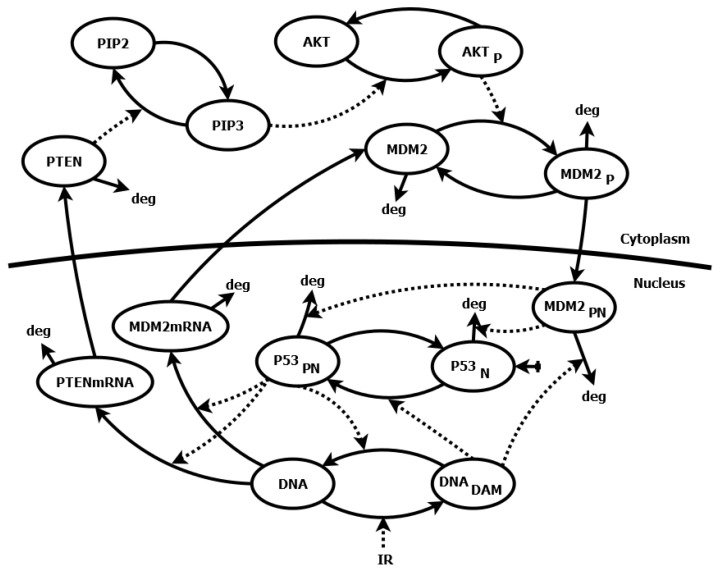
Schematic diagram of the p53 signaling pathway model [32]. The model involves two-compartmental kinetics of p53 protein, its primary inhibitor Mdm2, phosphatase PTEN, phosphatidylinositol 3-phosphate (PIP3) and Akt kinase and is activated by IR radiation, which leads to DNA damage ($DNA_{DAM}$). The N index stands for nuclear concentrations and P for phosphorylated molecules. Solid lines represent direct interactions in the model, such as production, degradation (deg) or state change (e.g., phosphorylated/nonphosphorylated) of selected molecules. Dashed lines represent indirect interactions, such as the catalysis of reactions or regulation of gene expression.

**Figure 2 ijms-23-06604-f002:**
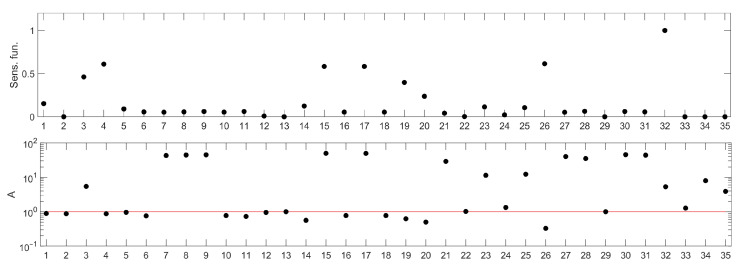
Parameter rankings for the p53 model: based on sensitivity functions (**upper panel**) and based on the method proposed in this paper (**lower panel**). The horizontal (*X*) axis represents the consecutive model parameters, whose names have been replaced by numbers for better readability. The parameter annotation associated with these numbers is given in Table 1. The red line in the lower panel corresponds to the *A* index value representing no influence of parameter change on the model response. Points located above this line represent parameters whose change (as described in the algorithm details) amplify the model response, while those below represent the parameters whose change lead to suppression of the model response.

**Figure 3 ijms-23-06604-f003:**
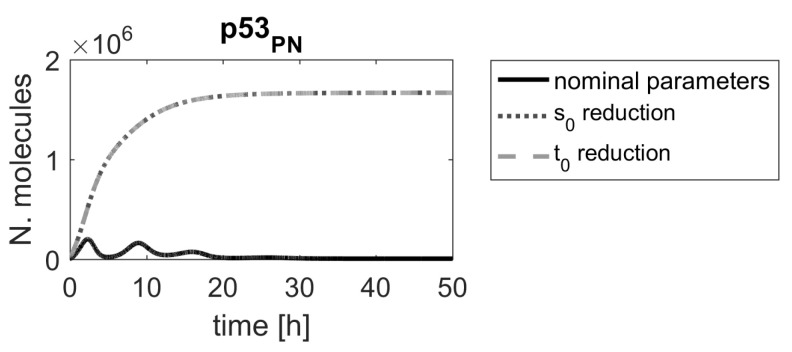
Comparison of p53 protein responses in the model with nominal parameters (black line) and parameters $s_{0}$, $t_{0}$ reduced by α=0.15 (gray dashed lines).

**Figure 4 ijms-23-06604-f004:**
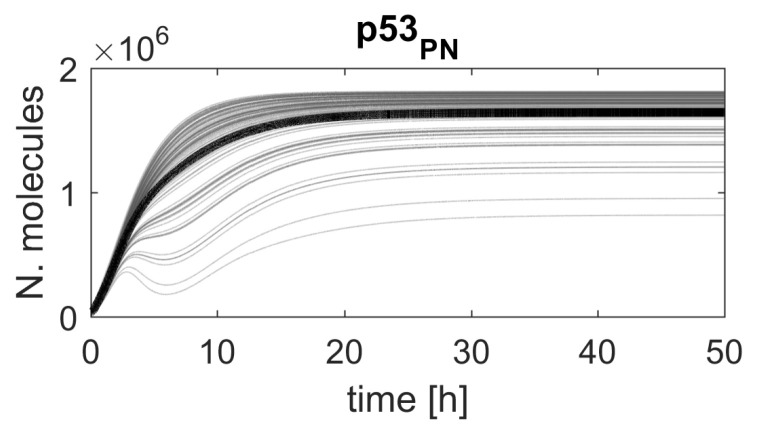
Simulated p53 protein responses in the model with parameter $t_{0}$ altered by the reduction factor α. The figure shows 100 randomly selected p53 protein responses (gray lines) and average p53 protein responses (black line) calculated from 1000 simulations.

**Figure 5 ijms-23-06604-f005:**
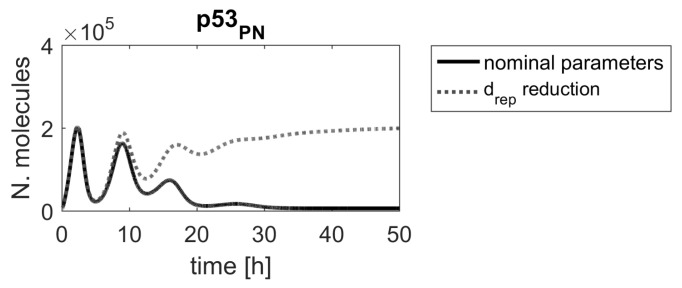
Comparison of p53 protein responses in the model with nominal parameters (black line) and parameter $d_{rep}$ reduced by α=0.15 (gray dashed line).

**Figure 6 ijms-23-06604-f006:**
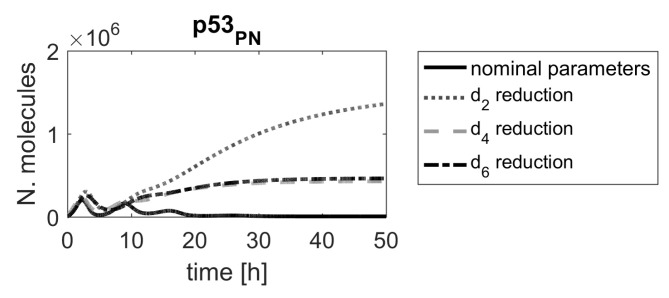
Comparison of p53 protein responses in the model with nominal parameters (black line) and parameters $d_{2}$, $d_{4}$ and $d_{6}$, reduced by α=0.15 (gray dashed line).

**Figure 7 ijms-23-06604-f007:**
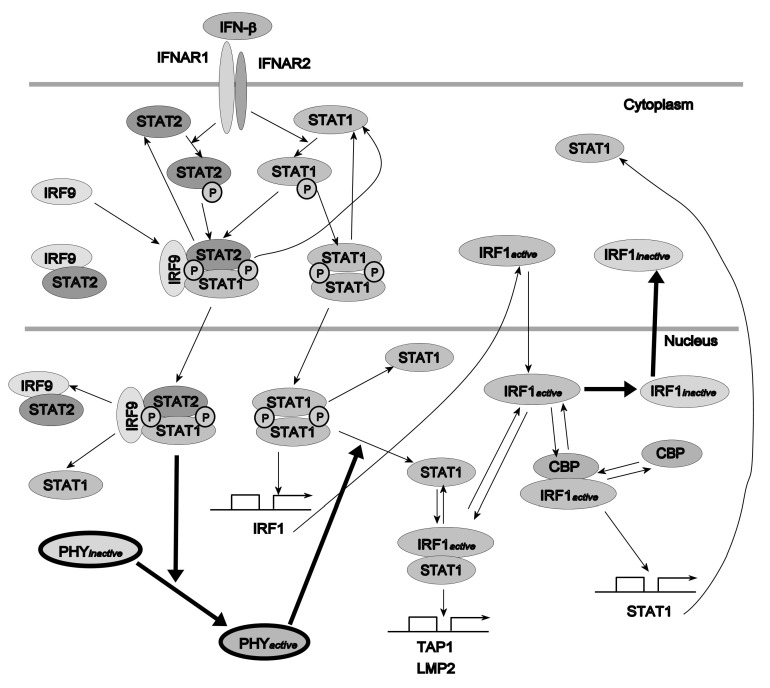
Schematic diagram of the IFN-β-induced JAK/STAT signaling pathway model. $PHY_{active}$ and $PHY_{inactive}$ represent unknown active and inactive phosphatases, respectively, hypothesized in the model [1].

**Figure 8 ijms-23-06604-f008:**
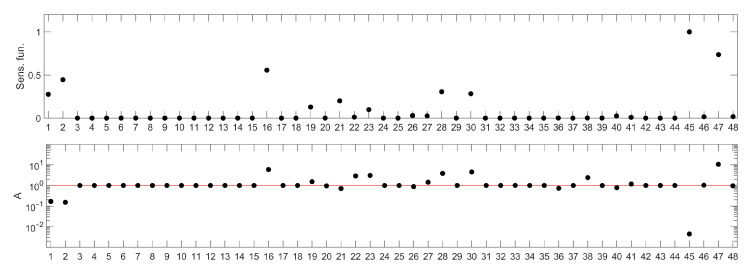
Parameter rankings for IFN-β-induced JAK/STAT signaling pathway model: based on sensitivity functions (**top panel**) and based on the method proposed in this paper (**bottom panel**). The horizontal (*X*) axis represents consecutive model parameters, whose names have been replaced by numbers for better readability. The parameter annotation associated with these numbers is given in Table 2. The red line in the lower panel corresponds to the *A* index value representing no influence of parameter changes on the model response. Points located above this line represent parameters whose change (as described in the algorithm details) amplify the model response, while those below represent the parameters whose change lead to the suppression of the model response.

**Figure 9 ijms-23-06604-f009:**
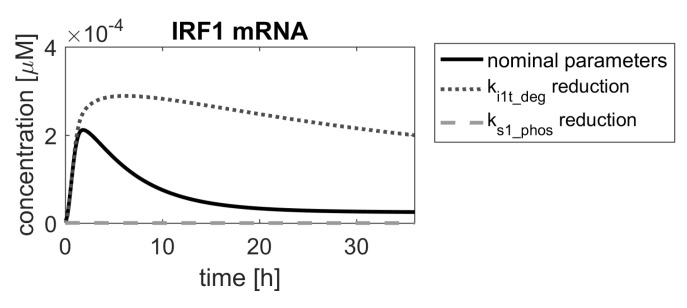
Comparison of IRF1 mRNA responses in the model with nominal parameters (solid black line) and parameters reduced by a factor of α=0.15 (gray dashed and dotted lines).

**Figure 10 ijms-23-06604-f010:**
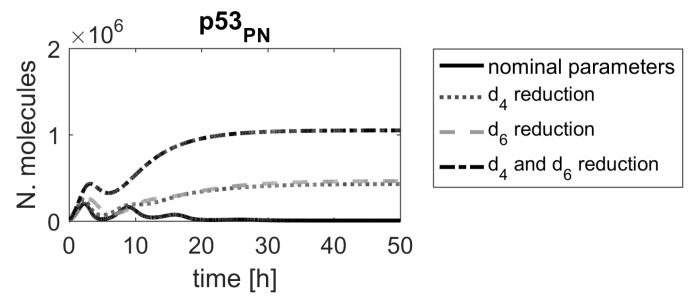
Comparison of p53 protein responses in the model with nominal parameters (black line) and parameters $d_{4}$ and $d_{6}$ reduced by α=0.15.

**Figure 11 ijms-23-06604-f011:**
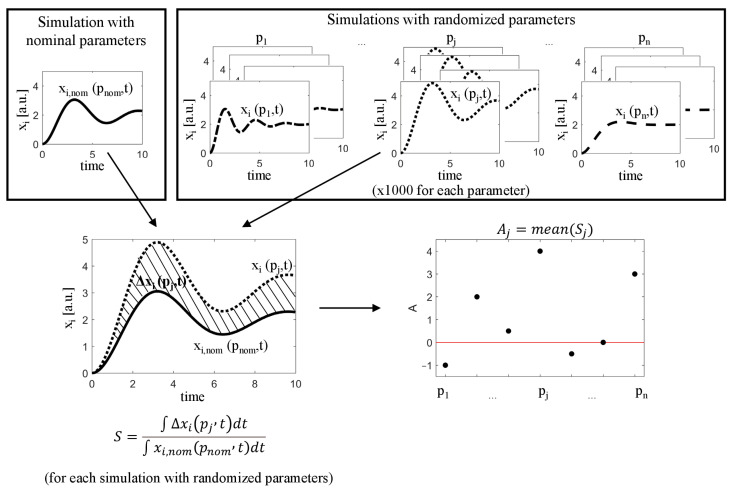
A simplified diagram showing the idea behind the algorithms used for sensitivity analysis.

**Figure 12 ijms-23-06604-f012:**
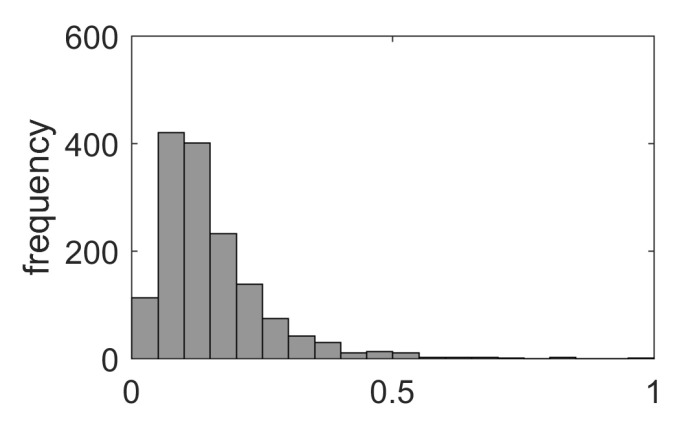
The distribution of reduction factor α.

**Table 1 ijms-23-06604-t001:** List of parameters appearing in the p53 signaling pathway model [32].

No.	Par.	Description	No.	Par.	Description
1	$a_{6}$	Max DNA damage rate	19	$d_{0}$	Mdm2 spontaneous deg. rate (all Mdm2 forms)
2	$q_{3}$	Coefficient governing apoptotic factor synthesis	20	$d_{1}$	DSB-induced Mdm2 deg. rate (all Mdm2 forms)
3	$d_{9}$	Apoptotic factors degradation rate	21	$d_{2}$	PTEN degradation rate
4	$p_{1}$	Max synthesis rate of apoptotic factor	22	$d_{3}$	Spontaneous $p53_{n}$ degradation rate
5	$a_{0}$	Spontaneous $p53_{n}$ phosphorylation rate	23	$d_{4}$	$Mdm2_{pn}$-induced $p53_{n}$ degradation rate
6	$a_{1}$	DSB-induced $p53_{n}$ phosphorylation rate	24	$d_{5}$	Spontaneous $p53_{pn}$ degradation rate
7	$a_{2}$	PIP activation rate	25	$d_{6}$	$Mdm2_{pn}$-induced $p53_{pn}$ degradation rate
8	$a_{3}$	AKT activation rate	26	$d_{7}$	$Mdm2_{t}$ degradation rate
9	$a_{4}$	Mdm2 phosphorylation rate	27	$d_{8}$	$PTEN_{t}$ degradation rate
10	$c_{0}$	$PIP_{p}$ dephosphorylation rate (by PTEN)	28	$i_{0}$	$Mdm2_{p}$ nuclear import
11	$c_{1}$	$AKT_{p}$ inactivation rate	29	$e_{0}$	$Mdm2_{pn}$ nuclear export
12	$c_{2}$	$Mdm2_{p}$ dephosphorylation rate	30	$AKT_{tot}$	Total number of Akt molecules (AKT+$AKT_{p}$)
13	$c_{3}$	Spontaneous $p53_{pn}$ dephosphorylation rate	31	$PIP_{tot}$	Total number of PIP molecules (PIP+$PIP_{p}$)
14	$p_{0}$	$p53_{n}$ production rate	32	$d_{rep}$	DNA repair rate
15	$s_{0}$	Mdm2 transcription rate	33	$q_{0}$	Spontaneous activation of Mdm2 and PTEN genes
16	$s_{1}$	PTEN transcription rate	34	$q_{1}$	$p53_{pn}$-depended activation of Mdm2 and PTEN genes
17	$t_{0}$	Mdm2 translation rate	35	$q_{2}$	Mdm2 and PTEN genes inactivation rate
18	$t_{1}$	PTEN translation rate			

**Table 2 ijms-23-06604-t002:** List of parameters appearing in the IFN-β-induced JAK/STAT signaling pathway model [1].

No.	Par.	Description	No.	Par.	Description
1	$k_{v}$	cytoplasmic/nuclear volume ratio	25	$k_{s1i1deg}$	[STAT1|IRF1] degradation rate
2	$v_{i1t}$	IRF1 transcription rate	26	$k_{s1s1}$	$\left[STAT1_{p}|STAT1_{p}\right]$ complex creation rate
3	$v_{s1t}$	STAT1 transcription rate	27	$k_{s1s2}$	$\left[STAT1_{p}|STAT2_{p}\right]$ complex creation rate
4	$v_{l2t}$	LMP2 transcription rate	28	$k_{phys1s1}$	$\left[PHY|STAT1_{p}|STAT1_{p}\right]$ complex creation rate
5	$v_{t1t}$	TAP1 transcription rate	29	$k_{s1i1}$	nuc. [IRF1|STAT1] complex creation rate
6	$k_{transl}$	translation rate	30	$k_{activation}$	PHY activation
7	*T*	time constant for inertial elements	31	$k_{inacti1}$	IRF1 inactivation rate
8	$k_{s1deg}$	STAT1 degradation rate	32	$k_{s1tprod}$	STAT1 constitutive mRNA production rate
9	$k_{s1pdeg}$	$STAT1_{p}$ degradation rate	33	$k_{s2tprod}$	STAT2 constitutive mRNA production rate
10	$k_{s2deg}$	STAT2 degradation rate	34	$k_{l2tprod}$	LMP2 constitutive mRNA production rate
11	$k_{s2pdeg}$	$STAT2_{p}$ degradation rate	35	$k_{t1tprod}$	TAP1 constitutive mRNA production rate
12	$k_{i1deg}$	$IRF1_{active}$ degradation rate	36	$e_{s1}$	STAT1 nuclear export
13	$k_{i1\_indeg}$	$IRF1_{inactive}$ degradation rate	37	$i_{s1}$	STAT1 nuclear import
14	$k_{s1t\_deg}$	STAT1 transcript degradation rate	38	$e_{s2}$	STAT2 nuclear export
15	$k_{s2t\_deg}$	STAT2 transcript degradation rate	39	$i_{s2}$	STAT2 nuclear import
16	$k_{i1t\_deg}$	IRF1 transcript degradation rate	40	$i_{s1s1}$	$\left[STAT1_{p}|STAT1_{p}\right]$ nuclear import
17	$k_{l2t\_deg}$	LMP2 transcript degradation rate	41	$i_{s1s2}$	$\left[STAT1_{p}|STAT2_{p}\right]$ nuclear import
18	$k_{t1t\_deg}$	TAP1 transcript degradation rate	42	$i_{i1}$	$IRF1_{active}$ nuclear import
19	$k_{inv\_s1s1}$	cyt. $\left[STAT1_{p}|STAT1_{p}\right]$ dissociation rate	43	$e_{i1}$	$IRF1_{active}$ nuclear export
20	$k_{inv\_s1s1\_n}$	nuc. $\left[STAT1_{p}|STAT1_{p}\right]$ dissociation rate	44	$e_{i1\_in}$	$IRF1_{inactive}$ nuclear export
21	$k_{inv\_s1s2}$	cyt. $\left[STAT1_{p}|STAT2_{p}\right]$ dissociation rate	45	$k_{s1\_phos}$	STAT1 phosphorylation rate
22	$k_{inv\_s1s2\_n}$	nuc. $\left[STAT1_{p}|STAT2_{p}\right]$ dissociation rate	46	$k_{s1\_dephc}$	$STAT1_{p}$ dephosphorylation rate
23	$k_{inv\_phys1s1}$	nuc. $\left[STAT1_{p}|STAT1_{p}|PHY\right]$ dissociation rate	47	$k_{s2\_phos}$	STAT2 phosphorylation rate
24	$k_{inv\_s1i1}$	nuc. $\left[STAT1|IRF1_{active}\right]$ dissociation rate	48	$k_{s2\_dephc}$	$STAT2_{p}$ dephosphorylation rate

## Data Availability

Not applicable.

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
