# Peer review of "Application of Sensitivity Analysis to Discover Potential Molecular Drug Targets"

_ijms, 2022, doi:10.3390/ijms23126604_

Round 1
Reviewer 1 Report
1. Please cite lines 43-52
2. Even if a drug stimulates a process, it acts through blocking the inhibitor of that process- this is a very generalized statement and not all drugs stimulate a target by inhibiting the blocking step, particularly in molecular mechanisms. For example, all recombinant protein directly binds to its target and stimulate the target protein- please reformat.
3. Figure 3: please explain the figure in legend and expand all abbreviations. What does a continuous and interrupted line mean in the figure? deg stands for?
4. Figure 4: Please include what X-axis annotates for? What does red line represent?
5. Line 226: it is not clear that the higher nuclear levels of p53 will be of wild type p53 or mutant p53 and how this model can distinguish between these two?
6. Lines 230-235: it is not clear from which figure and data this conclusion has been drawn (p7, p8, p21, p27, p30, p31 are associated with processes in the PTEN-PIP3-AKT loop).
7. Similar is the case with lines 236-239
8. Lines 282-293: not clear from which figure and data this conclusion has been drawn
9. Figure 9: The authors assume the pathway of IFN-b and stimulation of JAK-SATA pathway, however, JAK-STAT1/2 pathway is also stimulated by other IFNs like α, κ, δ, ε, τ, and ω. Please discuss how to apply this model in this context.
10. Please include “Limitations of the study” section after the conclusion section as this study is based on many assumptions and it is not necessary that in-silico findings will apply in biological in-vitro and in-vivo system. Additionally, there may be off-target effects, inhibiting one target may have other effects in biological system as it may have effect not only one but on other downstream signaling. Please discuss all limitations and emphasize the limitations of simulation and in-silico results.
11. Please check for typos.
Author Response
We would like to thank the reviewers for their valuable comments – indeed, our the previous manuscript contained phrases that were not entirely clear or could have been misleading.
The paper structure has been modified, to satisfy journal requirements, concerning sections order. All remaining changes have been recorded using MS Word – style formatting (as Tracking Changes is available only in Premium version of the Overleaf, to which we do not have access) – added text is in red, deleted text is in strikeout font. We have also provided a separate Supplement file with all model equations.
- Please cite lines 43-52
These lines (in the revised manuscript they are lines 46-61) express our strong opinion on the topic. We have reformulated the paragraph to make it clearer and, following the reviewer’s request, we have added two citations to support it.
- Even if a drug stimulates a process, it acts through blocking the inhibitor of that process- this is a very generalized statement and not all drugs stimulate a target by inhibiting the blocking step, particularly in molecular mechanisms. For example, all recombinant protein directly binds to its target and stimulate the target protein- please reformat.
Indeed, our phrasing was unfortunate, leading to the conclusion provided by the Reviewer. We have modified the text in the Introduction and Materials and Methods sections (lines 31-34 and 309-314 in the revised manuscript, respectively).
- Figure 3: please explain the figure in legend and expand all abbreviations. What does a continuous and interrupted line mean in the figure? deg stands for?
After changing the sections order to satisfy journal requirements, Fig. 3 became Fig. 1. We have expanded the figure caption, adding the meaning of the dashed line and abbreviations..
- Figure 4: Please include what X-axis annotates for? What does red line represent?
After changing the sections order to satisfy journal requirements, Fig. 4 became Fig. 2. The full explanation has been provided in the last paragraph of the Materials and Methods section. Moreover, we have modified and expanded the figure caption, explaining the annotation of the X axis and the meaning of the red line.
- Line 226: it is not clear that the higher nuclear levels of p53 will be of wild type p53 or mutant p53 and how this model can distinguish between these two?
The original model, referred to and utilized in this paper takes into account only one type of p53 and does not distinguish between wild type and mutant ones. The reviewer’s question addresses an important issue of what modelers actually describe by their equations. To answer that, however, a separate series of modeling and experimental research would be required. Mutated p53, if taken into account in the model, would call for completely different parameters, at the very least, if not a different system structure. In our opinion such investigation is beyond the scope of this paper. We simply assume that if the p53 works in the regulatory circuit as depicted in Fig. 1, its prolonged elevated levels lead to apoptosis,
- Lines 230-235: it is not clear from which figure and data this conclusion has been drawn (p7, p8, p21, p27, p30, p31 are associated with processes in the PTEN-PIP3-AKT loop).
After changing the sections order the remark refers to lines 118-122. We have modified the text as well as Table 1, adding the description columns and changed the text correspondingly.
- Similar is the case with lines 236-239
After changing the sections order the remark refers to lines 125-126. Similarly, as in the case of the previous remark, we have modified the text, adding the reference to the modified Table 1.
- Lines 282-293: not clear from which figure and data this conclusion has been drawn
After changing the sections order the remark refers to lines 173-180. We have modified the text, adding references to Table 2 and the bottom panel of Figure 8 and adding columns with parameter description to the Table 2.
- Figure 9: The authors assume the pathway of IFN-b and stimulation of JAK-SATA pathway, however, JAK-STAT1/2 pathway is also stimulated by other IFNs like α, κ, δ, ε, τ, and ω. Please discuss how to apply this model in this context.
The original model that has been utilized in this paper describes cellular responses to IFN-β stimulation. The same receptor is used by other type I IFNs (Moreover, type II IFN also activates STAT signaling). It is safe to assume that models describing system activation by other IFNs would have similar structure. However, they might require taking into account other STAT heterodimers (like e.g. STAT1|STAT3) and reestimation of nominal parameter values. Another interesting issue is SOCS-based regulation, which seems to be cell-type dependent. The experimental results on which the original model was based showed no SOCS-1 induction in HeLa cells, though, in general, SOCS-1 provides the negative feedback to IFN-β cellular responses (it is mentioned in the text). Therefore, analysis of another IFN-induced system can be performed with the method presented in the paper (it might even lead to similar conclusions) but should be preceded by experiment-based model fitting.
- Please include “Limitations of the study” section after the conclusion section as this study is based on many assumptions and it is not necessary that in-silico findings will apply in biological in-vitro and in-vivo system. Additionally, there may be off-target effects, inhibiting one target may have other effects in biological system as it may have effect not only one but on other downstream signaling. Please discuss all limitations and emphasize the limitations of simulation and in-silico results.
We couldn’t agree more with this remark. We have added a separate paragraph in the Conclusions section (as the journal require a particular section structure}.
- Please check for typos.
We have carefully read the manuscript and corrected typing errors that we have found.
Reviewer 2 Report
This manuscript presents a novel method of sensitivity analysis which aims to find potential molecular drug targets within intracellular signalling pathways. The feasibility of this method has been shown in two different signalling pathways (p52 and JAK/STAT). It is a well written manuscript with clearly presented methodology and results. I have no specific comments.
Author Response
We would like to thank the reviewers for their valuable comments – indeed, our the previous manuscript contained phrases that were not entirely clear or could have been misleading.
The paper structure has been modified, to satisfy journal requirements, concerning sections order. All remaining changes have been recorded using MS Word – style formatting (as Tracking Changes is available only in Premium version of the Overleaf, to which we do not have access) – added text is in red, deleted text is in strikeout font. We have also provided a separate Supplement file with all model equations.
Round 2
Reviewer 1 Report
None